RESEARCH

# Partial gene suppression improves identification of cancer vulnerabilities when CRISPR-Cas9 knockout is pan-lethal

J. Michael Krill-Burger[1], Joshua M. Dempster[1], Ashir A. Borah[1], Brenton R. Paolella[1], David E. Root[1], Todd R. Golub[1,2,3], Jesse S. Boehm[1], William C. Hahn[1,2,3], James M. McFarland[1], Francisca Vazquez[1,2]* and Aviad Tsherniak[1]

*Correspondence:
vazquez@broadinstitute.org

[1] Broad Institute of Harvard and MIT, Cambridge, MA, USA
[2] Dana-Farber Cancer Institute, Boston, MA, USA
[3] Harvard Medical School, Boston, MA, USA

## Abstract

**Background:** Hundreds of functional genomic screens have been performed across a diverse set of cancer contexts, as part of efforts such as the Cancer Dependency Map, to identify gene dependencies—genes whose loss of function reduces cell viability or fitness. Recently, large-scale screening efforts have shifted from RNAi to CRISPR-Cas9, due to superior efficacy and specificity. However, many effective oncology drugs only partially inhibit their protein targets, leading us to question whether partial suppression of genes using RNAi could reveal cancer vulnerabilities that are missed by complete knockout using CRISPR-Cas9. Here, we compare CRISPR-Cas9 and RNAi dependency profiles of genes across approximately 400 matched cancer cell lines.

**Results:** We find that CRISPR screens accurately identify more gene dependencies per cell line, but the majority of each cell line's dependencies are part of a set of 1867 genes that are shared dependencies across the entire collection (pan-lethals). While RNAi knockdown of about 30% of these genes is also pan-lethal, approximately 50% have selective dependency patterns across cell lines, suggesting they could still be cancer vulnerabilities. The accuracy of the unique RNAi selectivity is supported by associations to multi-omics profiles, drug sensitivity, and other expected co-dependencies.

**Conclusions:** Incorporating RNAi data for genes that are pan-lethal knockouts facilitates the discovery of a wider range of gene targets than could be detected using the CRISPR dataset alone. This can aid in the interpretation of contrasting results obtained from CRISPR and RNAi screens and reinforce the importance of partial gene suppression methods in building a cancer dependency map.

**Keywords:** Functional genomics, Cancer genomics, Target identification, RNAi, CRISPR-Cas systems, CRISPR-Cas9 genome editing, Drug discovery

## Background

The Cancer Dependency Map Project (DepMap) aims to accelerate cancer precision medicine by systematically identifying cancer vulnerabilities through genetic [1–4] and small-molecule perturbation screens [5–8]. Genetic loss-of-function screens measure cell viability in response to perturbation of individual gene function (negative selection) and can be applied at a genome-wide scale. Understanding the differences between genetic screening technologies, such as RNAi and CRISPR-Cas9, is important for the interpretation of analyses within the DepMap framework.

Initially, the development of assays based on RNA interference (RNAi) enabled parallel screening across many cell types by synthesizing large pooled reagent libraries, and led to the identification of numerous prospective therapeutic targets [9–14]. Unfortunately, a limitation of RNAi is that reagent sequences frequently overlap with miRNA's seed sequences, resulting in modulation of unintended targets (off-target effects). This often resulted in poor data reproducibility and was proposed as a contributing factor to low success rates in clinical trials [15]. Computational modeling of off-target seed effects, performed on the DepMap RNAi datasets using DEMETER2 [16], substantially improved the on-target gene effect estimates. However, this method may not be practical for small experiments, leaving an opportunity for new and improved genome engineering technologies.

In recent years, cell viability screening with CRISPR-Cas systems has increased due to superior efficacy and specificity. For example, CRISPR-Cas9 screens performed at the Broad and Sanger Institutes identified highly concordant common and selective dependencies as well as robust biomarkers [17]. However, it's unclear whether large CRISPR-Cas9 datasets should be viewed as a replacement for RNAi datasets considering the different mechanisms by which the technologies inhibit gene function. RNAi reagents (shRNAs) directly bind target mRNAs and cause their degradation, but typically result in partial depletion of target mRNA expression [18, 19]. CRISPR-Cas9 reagents (sgRNAs) induce DNA double-strand breaks, which have the potential to produce a true null phenotype through frameshift mutations, but often result in a heterozygous mixture of cells due to protein-conserving mutations or alternative splicing. Whether the cellular viability effect of RNAi knockdown or CRISPR knockout is the most relevant measurement could depend on the research question. For example, identifying a comprehensive list of dependencies per cell line might benefit from one approach while identifying potential cancer vulnerabilities by differential dependency across cell lines might benefit from a different approach.

Previous comparisons of CRISPR-Cas9 and RNAi screens have focused on the agreement between genetic dependencies discovered per cell line [20, 21], but have not investigated how the method of gene inhibition impacts the patterns of gene dependency across cell lines. Since a goal of DepMap is to enable researchers to investigate mechanisms of genetic dependency, it is important to know whether CRISPR or RNAi datasets show patterns of dependency that better match cellular genomic features or chemical compound sensitivity. Now with 403 shared cell lines between the DepMap RNAi and CRISPR datasets, it is possible to perform a side-by-side comparison of analyses that integrate large-scale genomic or drug sensitivity datasets. Here we explore how the choice of perturbation type affects the classification of dependencies as common or

selective across cell lines and the downstream implications for cancer target discovery and gene function inference.

## Results

### Selecting data to represent each perturbation type

The Cancer Dependency Map Project provides a collection of loss-of-function screens from several sources, including two large-scale RNAi experiments (Achilles [1], Project DRIVE [4]) and two genome-wide CRISPR experiments (DepMap [2], Project SCORE [3]). To help select data that best represents each perturbation type (CRISPR, RNAi), we benchmarked the quality of each cell line screen, using both unprocessed data (individual reagents or mean of reagents targeting the same gene) and processed data (gene effect estimates from CERES [2] or DEMETER2 [16]). As expected, the CRISPR reagent-level data had superior screen quality metrics compared to RNAi (Additional file 1: Fig. S1). However, the margin of improvement was narrowed when using processed datasets (Additional file 1: Fig. S2a). Data processing also improved the agreement across perturbation types (correlation between pairs of RNAi and CRISPR datasets), which suggests the processing methods reduced perturbation-specific artifacts (Additional file 1: Fig. S2b-c). Therefore, we chose to use processed data for comparison of CRISPR and RNAi, and selected the DepMap [22] (CRISPR) and DEMETER2-Combined [16] (RNAi) datasets to maximize the number of shared cell lines ($N = 403$).

Additional gene filtering was applied to CRISPR and RNAi datasets, where indicated, to increase the likelihood that differences observed between the chosen datasets will generalize to other datasets as well. To characterize the landscape of dependencies observed in each individual dataset, we included all genes shared between RNAi and CRISPR datasets ($N = 15{,}221$). In cases where methods could be sensitive to the number of missing values or reagents per gene, we repeated analyses using only genes included in the Project DRIVE library (6901 genes). This increased the average number of RNAi reagents per gene from 6 to 17 and decreased the missing values from ~20 to ~5%. Finally, when directly comparing dependencies between CRISPR and RNAi datasets, which is sensitive to individual outliers caused by differences in experimental design, such as screen duration or cell culture media, we restricted the analysis to a set of high-confidence genes ($N = 1703$) with agreement between pairs of datasets of the same perturbation type (Additional file 1: Fig. S3, Additional file 2: Table S1, "Methods"). Regardless of the gene filtering method, CRISPR and RNAi datasets always have identical cell lines and genes, and any missing values are removed from the other datasets to match. This approach allows us to maintain the simplicity of pairwise comparisons between perturbation types while reducing outliers driven by experiment-specific conditions.

### Dependencies identified using each perturbation type

Gene *dependency* represents a decrease in cellular viability following gene perturbation. The phenotype is measured by a change in cell count 14–28 days after perturbation, which is sensitive to gene perturbations that kill cells ("cytotoxic") as well as perturbations that inhibit cell growth or proliferation ("cytostatic"). This results in a continuous distribution of dependency values for each gene, called gene effects when using processed data, where a more negative gene effect indicates stronger dependency. We refer

to a gene that is deemed a dependency in at least 90% of cell lines, as a *pan-dependency*. We use the term *cell essential genes* to refer to a theoretical set of genes that are required for cell survival. While the results of pan-dependency analysis (Additional file 1: Fig. S4) might help identify cell essential genes, the meaning of gene inclusion in this set is conceptually different.

Several factors affect the ability to detect dependencies when perturbing genes using CRISPR or RNAi, including the strength of on-target effects ("efficacy"), as well as the pervasiveness of off-target effects ("specificity"). We observed that dependencies detected using CRISPR were nearly a superset of those detected using RNAi (Fig. 1a), confirming similar observations from previous studies [20, 21, 23]. To determine whether the increased number of CRISPR dependencies is driven by improved efficacy or specificity, we compared the performance of positive and negative control gene sets. We found that when using reagent-level datasets (Fig. 1b, Additional file 1: Fig. S5a), CRISPR specificity was superior to RNAi, as indicated by the lower variance in the fold-change values of negative controls (non-expressed or nonessential genes [24]). However, when comparing processed gene-level estimates (generated by CERES, DEMETER2), we found that RNAi specificity was similar to CRISPR (Fig. 1c, Additional file 1: Fig. S5b) while maintaining an equivalent dynamic range with respect to RNAi positive controls (core essential genes [24] identified from other RNAi screens). Instead, the major difference between processed datasets was an increase in CRISPR efficacy (Fig. 1c) across a larger set of positive controls (unbiased essential genes derived from gene trap experiments [25], embryonic lethal mouse genes [26], and studies of intolerance to germline loss-of-function in human populations [27–32], "Methods"). Therefore, the processed RNAi dataset shows similar specificity to CRISPR, likely a testament to the efforts put into screening multiple reagents per gene and developing computational methods to

(See figure on next page.)

**Fig. 1** CRISPR knockout produces stronger viability effect than RNAi knockdown. **a** Mean gene effect across 403 cell lines for 15,221 genes overlapping the CRISPR and RNAi datasets. Each axis is divided into 100 bins (2D histogram) and the color of the dots represents the number of data points that fall into each bin. **b** $Log_2$ fold-change (LFC) values for reagents targeting the same gene are collapsed to gene means per cell line. Boxes represent the distribution of gene mean LFC across all cell lines for genes included in the nonessential control set [24] (738 genes), non-expressed set (10 randomly sampled genes from each cell line with RNAseq $log_2(TPM + 1) < .2$), unbiased essential gene set described in the "Methods" (744 genes), or the core essential control set [24] (208 genes). CRISPR reagents are from the Avana library and RNAi reagents are those used in the DEMETER2-combined [16] RNAi dataset (union of Achilles and DRIVE libraries). Whiskers are extended to 4 times the interquartile range to limit overplotting outliers since over 1.2 million data points are represented. **c** Unscaled gene effect estimates from CERES (CRISPR) and DEMETER2 (RNAi) were normalized per cell line (*Z*-score) and grouped into the same control sets and plot parameters from part *b*. **d** Each gene ($N = 15,221$) is included in at least one dependency class. Density plots illustrate the difference in dependency patterns between dependency classes and represent the union of CRISPR gene effects for all genes in each dependency class. The horizontal bars represent the number of genes identified as members of the respective dependency classes. **e** Mapping between the disjoint CRISPR and RNAi dependency class for each high-confidence gene ($N = 1703$). **f** Each cell line ($N = 403$) is a stacked bar where bar segments represent the proportion of gene dependencies relative to the total number of genes profiled in each cell line (*y* axis). Labels indicate the mean percent of dependencies per cell line that are classified as pan-dependencies in the overall CRISPR or RNAi dataset. Colors correspond to the gene dependency classifications as indicated in part *d*. **g** Intersection of pan-dependencies identified using CRISPR and RNAi datasets with essential genes identified using ExAC, mouse knockout, and gene trap. **h** Ability to separate genes classified as pan-dependencies in both CRISPR datasets ($N = 1339$) from genes that were not pan-dependencies in either CRISPR dataset ($N = 14,702$) using ROC AUC of mean gene trap results for KBM7 and HAP1 (AUC = 0.96), RNAi mean D2-Combined gene effect (AUC = 0.86), and the median of ExAC constrained loss-of-function metrics (AUC = 0.65)

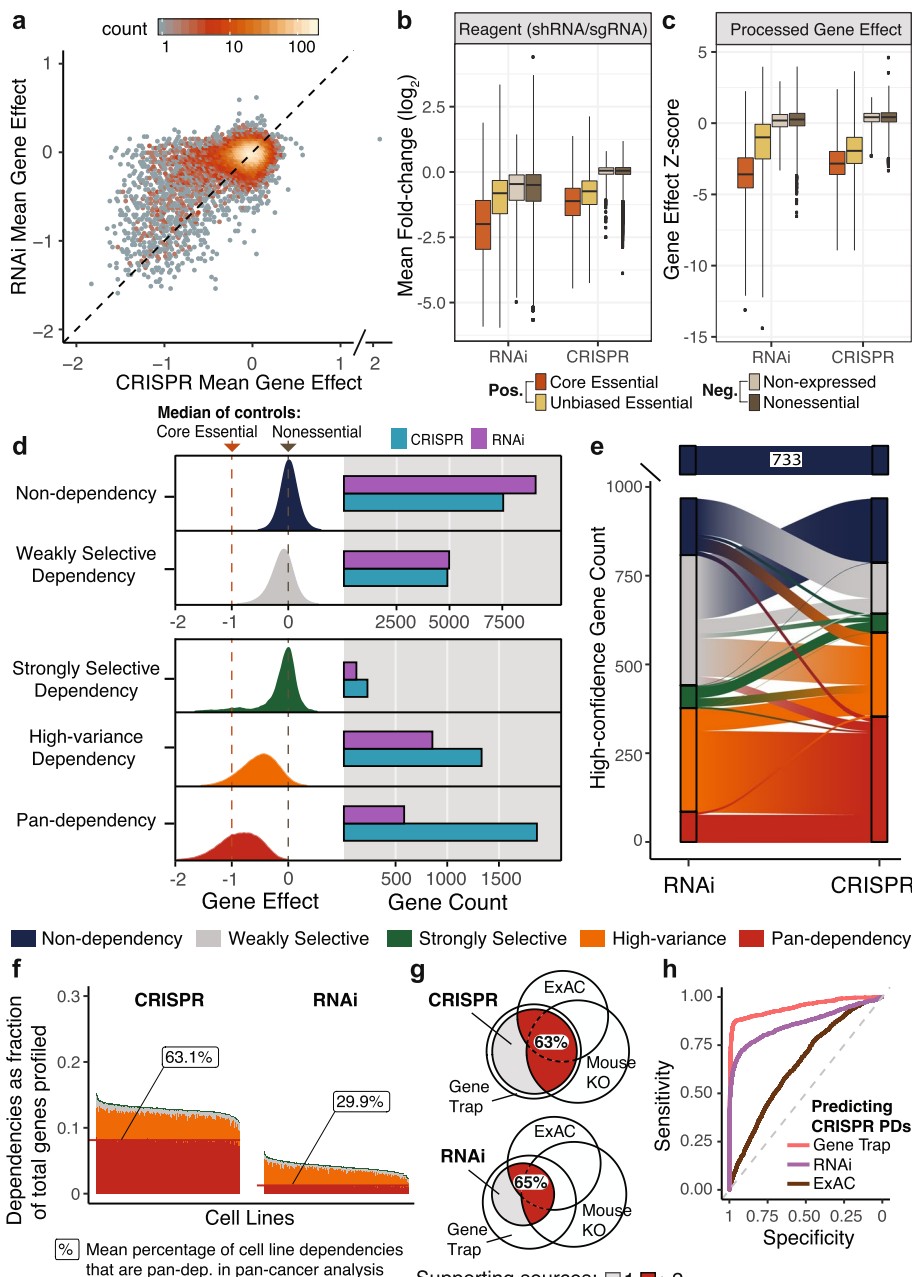

**Fig. 1** (See legend on previous page.)

address reagent off-target effects [1, 4, 16], but the CRISPR dataset shows stronger efficacy across a broader set of gene targets.

In addition to simply detecting dependencies, we would like to identify genes that show patterns of selective dependency across cell lines. A selective pattern suggests that particular cancer models are more sensitive than others to the inhibition of a gene, which could represent a targetable cancer vulnerability. To evaluate how the type of perturbation impacts our ability to identify dependency patterns, we defined dependency classes (Fig. 1d, "Methods," Additional file 3: Table S2) that represent three selective patterns (high-variance, strongly selective (Additional file 1: Fig. S6), weakly selective) and

two uniform patterns, meaning dependencies that are common to most cell lines (pan-dependent) or no cell lines (non-dependent). Multiple classes were required to capture selective dependencies because their gene effect distributions are often shaped quite differently; for example, BRAF dependency is strongly selective while ADAR dependency is high-variance (Additional file 1: Fig. S7). Comparing the number of genes identified per dependency class, we found the largest difference between perturbation types was a threefold increase in pan-dependencies (1867 total genes) identified using CRISPR compared to RNAi (Fig. 1d). When performing hit calling per cell line (genes with greater than 50% probability of dependency, "Methods"), the increase in CRISPR pan-dependencies translates to an average of 63% of dependencies detected per cell line being pan-dependencies using CRISPR as opposed to 30% using RNAi (Fig. 1f, Additional file 1: Fig. S8). While these results are consistent with prior observations that more dependencies are detected using CRISPR knockout, it was not necessarily expected that the majority of new CRISPR-specific dependencies would be pan-dependencies.

Given the large set of CRISPR pan-dependencies, we wanted to confirm that the CRISPR-specific pan-dependencies share similar agreement with essential genes identified through alternative methodologies (gene trap [25], mouse knockout lethality [26], constraint of deleterious mutations in ExAC [27–32]) so as to suggest they are not artifacts of the CRISPR-Cas9 mechanism. We found that CRISPR and RNAi pan-dependencies were confirmed by the alternative methodologies in similar proportions (Fig. 1g), but enriched for different functional groups (Additional file 1: Fig. S9-S11, Additional file 1: Supplemental Notes). Additionally, we found that gene trap data (knockout through mutagenesis in cell lines) was the most predictive of CRISPR pan-dependency status (ROC AUC = 0.96, Fig. 1h), suggesting that CRISPR-specific pan-dependencies are genes that require complete knockout to produce a consistently strong effect on cell viability.

Since more selective dependencies were also identified using CRISPR, including a 1.88 fold increase in strongly selective dependencies and a 1.56 fold increase in high-variance dependencies (Fig. 1d), we questioned whether the RNAi dataset was providing any complementary information. To answer this question, we mapped each gene between its RNAi and CRISPR dependency class (Fig. 1f, "Methods") using high-confidence dependencies ($N = 1703$) to minimize experiment-specific outliers. We found that the CRISPR dataset is sufficient for identifying pan-dependencies since 99.6% of RNAi pan-dependencies were also CRISPR pan-dependencies (Fig. 1f, Additional file 1: Fig. S12a). However, each perturbation type had more distinct than shared selective dependencies (Additional file 1: Fig. S12b, c), the bulk of the misaligned genes were CRISPR pan-dependencies that were selective using RNAi or CRISPR selective dependencies that were non-dependent using RNAi (Fig. 1f). This suggests that stronger CRISPR efficacy results in the discovery of new selective dependencies that were missed by RNAi, but also results in decreased selectivity among CRISPR pan-dependencies, which typically have higher variance using RNAi.

Based on these observations, we expect individual dependencies (a specified gene and cell line) detected using RNAi to be corroborated by the higher efficacy CRISPR knockout, but we do not necessarily expect the CRISPR dataset to reproduce the same selectivity pattern across cell lines that is observed using RNAi. For example, only 25.6%

of the genes classified as high-variance dependencies using RNAi were also classified as high-variance using CRISPR (Additional file 1: Fig. S12), but 99.2% of the cell lines called dependent on any of these distinct RNAi high-variance genes were supported by the dependency calls using CRISPR (Additional file 1: Fig. S13). This occurs because the majority of high-variance dependencies using RNAi are pan-dependencies using CRISPR, likely a result of different responses to knockdown and knockout. Whether the larger dynamic range in RNAi gene effects helps to uncover biological relationships that could be missed for genes that are pan-dependencies using CRISPR is assessed through the following analyses of predictive omics markers, drug-gene target correlations, and co-dependencies.

### Biomarker identification for therapeutic target discovery

One of the critical steps of current target discovery efforts is identifying molecular features ("biomarkers") that could provide insights into the mechanism underlying the cellular dependency. Using biomarkers to guide preclinical target discovery is motivated by the increased success of phase II and III clinical trials for drugs with genetic evidence linking the mechanism of action to disease biology [33]. Therefore, we wanted to know if the analysis of RNAi or CRISPR datasets leads to the identification of more or better biomarkers. Here we determine which dependencies have accurate predictive models and then evaluate the individual predictive features of each model to characterize the different types of potential biomarkers.

We trained multivariate regression models (random forest) to predict RNAi or CRISPR gene effects for each gene using multi-omics features, most notably RNAseq-derived gene expression, gene-level copy number, and mutation calls ("Methods," Additional file 4: Table S3). Cell lines and missing values were matched between dependency datasets to remove sample bias. To benchmark the predictive modeling performance, we evaluated the accuracy (correlation between measured and predicted gene effect profiles) for a set of clinically actionable oncology targets [34]. We found that actionable targets are among the best-predicted gene dependencies (median > 97th percentile) for both perturbation types, with similar performance for KRAS, NRAS, BRAF, ESR1, PIK3CA, and EGFR (Additional file 1: Fig. S14a, b). Although there is one actionable target, MET, that is better predicted using CRISPR data, these results indicate that our approach to predictive modeling is applicable to both perturbation types.

Evaluating the accuracy of CRISPR and RNAi predictive models for each gene dependency profile ($N = 15{,}221$), we found there were more accurate predictive models ($r > 0.5$) for CRISPR dependencies than RNAi, unless the target gene was a CRISPR pan-dependency (Fig. 2a). More specifically, the mean accuracy of RNAi predictive models surpassed CRISPR models when over ~75% of cell lines were dependent using CRISPR (Fig. 2b). To assess the biological relevance of the features used in the accurate predictive models, we determined the most important individual predictive feature (Gini importance) of each multivariate model and labeled it a "related gene" if it is an omics measurement of an individual gene with prior evidence to support its associated with the target gene ("Methods"). We found that related genes were used as top features for 51% of CRISPR models and 29% of RNAi models, though this increased to 45% of RNAi models when focusing on the subset of genes included in Project DRIVE that were

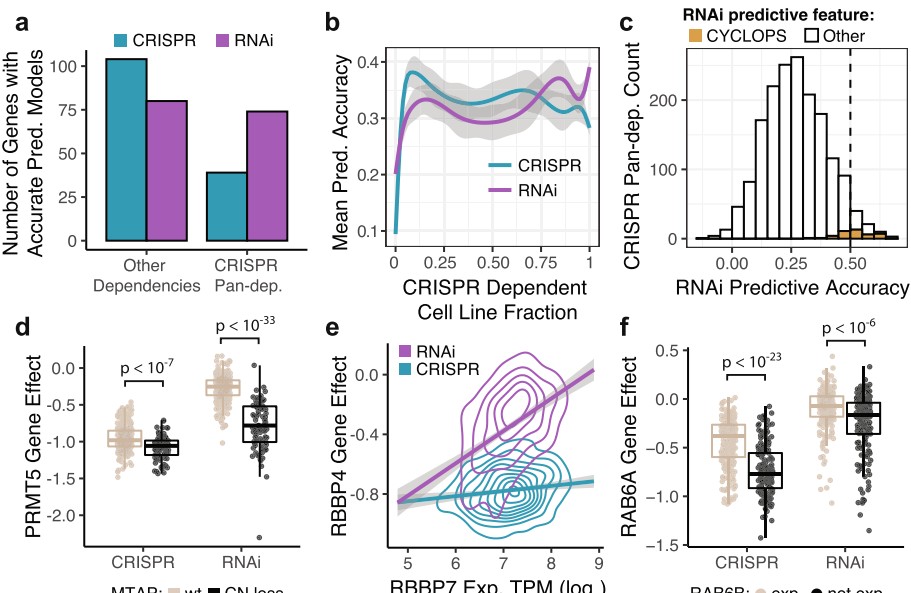

**Fig. 2** Predictive models for dependency. **a** Multivariate regression of each CRISPR or RNAi gene effect profile ($N = 15{,}221$) using predictive features derived from omics datasets and cell line annotations. The accuracy of each predictive model is the correlation coefficient of measured and predicted values across cell lines. The number of accurate predictive models ($r > 0.5$) is split according to whether the gene target is also a CRISPR pan-dependency. **b** Mean predictive accuracy for high-confidence dependencies ($N = 1703$) as a function of the fraction of cell lines identified as dependencies using CRISPR (probability of dependency $> 0.5$). **c** CYCLOPS genes have a positive correlation ($r > 0.5$) between RNAi gene effects and copy number and account for ~35% of the accurate RNAi models for CRISPR pan-dependencies ($N = 1719$) after removing 148 RNAi models driven by confounding factors. **d** Genetic perturbation of PRMT5 using RNAi results in larger effect size between cell lines with MTAP copy number loss and MTAP wild-type compared to CRISPR knockout (Wilcoxon *p*-value: CRISPR $= 4.6 \times 10^{-8}$, RNAi $= 2.5 \times 10^{-34}$). **e** RNAi gene effect of RBBP4 is more correlated with expression of paralog gene RBBP7. Density (2D) contours represent 402 cell lines for each genetic dependency dataset (RNAi, CRISPR). **f** CRISPR knockout of RAB6A has a more negative viability effect on cells lacking expression of paralog gene RAB6B (Wilcoxon *p*-value: CRISPR $= 6.2 \times 10^{-24}$, RNAi $= 6.1 \times 10^{-7}$)

targeted with more reagents per gene (Additional file 1: Fig. S14c). We also found that RNAi predictive models for CRISPR pan-dependencies were more likely to use related genes as top features compared to RNAi models for all other dependencies (Additional file 1: Fig. S14c). This means that on average, CRISPR dependency profiles are better explained by related omics features, but also suggests that RNAi dependency profiles for CRISPR pan-dependencies could be therapeutically relevant since there are often biologically relevant markers that predict increased sensitivity to knockdown by RNAi.

Deeper investigation of the relationship between features used to predict a gene dependency can aid in generating hypotheses about the mechanism of the dependency. Several types of relationships between genetic dependencies and predictive features ("biomarker classes") have been commonly observed, including genetic driver, expression addiction, paralog, and CYCLOPS [1, 4, 35, 36]. We evaluated the features used by each predictive model and, based on our definitions for the biomarker classes ("Methods"), assigned 80 CRISPR models and 76 RNAi models to at least one class (Additional file 1: Fig. S15a). We found that more CRISPR models were classified as expression addiction (dependency predicted by expression of the target gene) or paralog (dependency predicted by omics features of the target gene's paralogs), and a greater number of

models were classified as CYCLOPS models (stronger dependency associated with lower copy number of the target gene) using RNAi (Additional file 1: Fig. S15a).

Since CYCLOPS dependencies represent essential genes that are relatively more sensitive to inhibition due to copy number loss, we questioned whether all of the accurate RNAi models for CRISPR pan-dependencies would be considered CYCLOPS. We found that approximately 35% of accurate RNAi models were CYCLOPS (Fig. 2c, Additional file 1: Fig. S15b), which leaves potential for RNAi models to identify other types of synthetic lethal or novel dependency-biomarker relationships. For example, PRMT5 and RBBP4 are CRISPR pan-dependencies that are better predicted by their respective biomarkers, MTAP copy number [14] (Fig. 2d) and RBBP7 expression (Fig. 2e), using RNAi. For other CRISPR pan-dependencies, such as RAB6A, the CRISPR signal is not completely saturated and association of the dependency with low expression of its paralog RAB6B can be detected using either perturbation type (Fig. 2f).

### Associations between genetic and chemical screens

Similarity in cell line response to genetic perturbation and targeted small-molecule drug treatment can be used to generate hypotheses about drug targets or mechanism of action. However, we do not expect all drug sensitivity profiles to be highly correlated with the dependency profiles of their annotated targets due to drug polypharmacology and incomplete target annotations. Therefore, we used three large drug screening datasets with different experimental designs (GDSC [37, 38], CTD2 [6, 39], PRISM [5]) to help identify subtle but consistent effects. The same normalization method was applied to each dose-level drug dataset ("Methods"), and each CRISPR-RNAi comparison includes drugs that were strongly correlated to an annotated gene target using at least one genetic perturbation type.

First, we questioned whether the CRISPR and RNAi profiles for each drug target are better correlated to different concentrations of the drug. Using genes included in the Project DRIVE RNAi library and CTD2 drugs with annotated protein targets [40], we found that concentrations towards the center of the 16 dose range, which are expected to have the most selective drug responses by design, were most likely to have an annotated target as a top dependency correlate regardless of genetic perturbation type (Fig. 3a). We also found that for each drug target, the CRISPR and RNAi dependency profiles typically correlate best to the same drug dose (Fig. 3b). The results were similar for the 8 concentrations tested for drugs included in the GDSC and PRISM datasets (Additional file 1: Fig. S16). This suggests that the differences between drug doses are typically larger than the differences between RNAi and CRISPR dependency data for annotated targets.

Next, we evaluated whether correlating drug sensitivity profiles to RNAi or CRISPR datasets recovered more annotated drug targets. Using the best correlated dose to represent each drug, we found RNAi and CRISPR profiles had similar mean drug correlation for annotated targets (Fig. 3c). Among the strongest overall correlations were drugs targeting BRAF, EGFR, IGF1R, and ERBB2, which had similar performance using RNAi or CRISPR data (Fig. 3d). However, there were several drugs that correlated better with target dependency using RNAi (FLT3, BCL2L1, FGFR2, CHEK1, WEE1, CDK2, and AURKA) and others using CRISPR (MAPK14, JAK2, ABL1, RARA, PI4KB, NAE1) (Fig. 3d). Evaluating the targets better correlated using RNAi, we observed that the

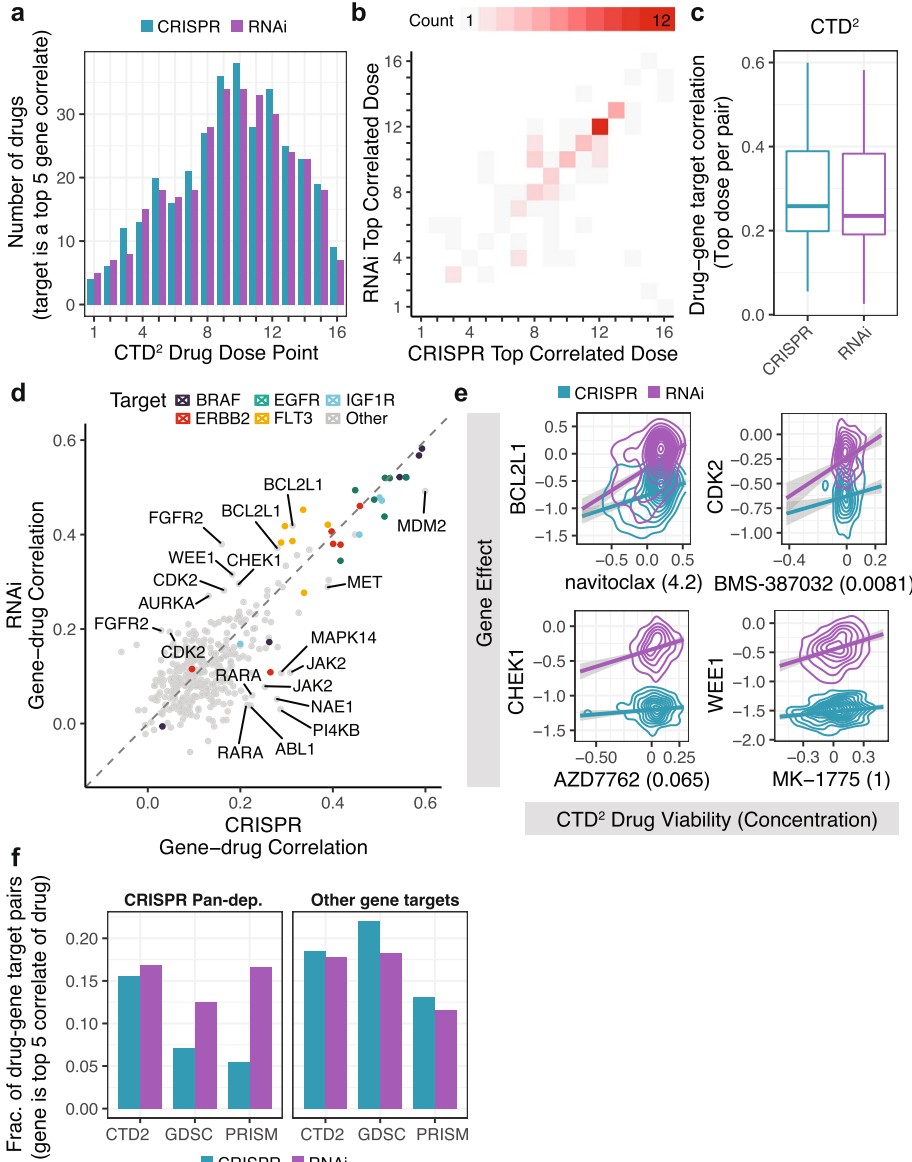

**Fig. 3** Associations between drug sensitivity and drug-target dependency. **a** Number of CTD2 drug sensitivity profiles that have an annotated target within the top 5 correlated gene dependencies per drug dose. There are 165 drugs included that use the standard 16-point concentration range. **b** Maximum correlated drug dose for each annotated drug-gene target pair ($N=86$) using CRISPR compared to RNAi. Drug-gene target pairs are included if the target is among the drug's top 5 gene correlates using either CRISPR or RNAi. **c** Pearson correlation of drugs and annotated targets ($N=88$) as in part *b* except without removing pairs for non-standard concentration ranges. **d** Correlation of each CTD2 drug with its annotated gene targets in the CRISPR and RNAi datasets (167 drugs, 375 gene targets). **e** Relationship between gene effect and CTD2 drug sensitivity for 4 drugs and their annotated gene targets (BCL2L1, CHEK1, CDK2, WEE1) which are CRISPR pan-dependencies. Density (2D) represents over 300 cell lines for each genetic dependency dataset (RNAi, CRISPR) per drug. Data is smoothed using linear models with 95% confidence intervals. **f** Fraction of annotated drug-target pairs where the target gene is among the drug's top 5 most correlated gene dependencies. Fractions are calculated per drug dataset (CTD2, GDSC, PRISM) since each dataset includes different drugs and different totals of drug-gene target pairs. Pairs are faceted by whether the target gene is a pan-dependency using CRISPR

majority (BCL2L1, CHEK1, WEE1, CDK2, and AURKA) were CRISPR pan-dependencies and had greater variance using RNAi (Fig. 3e). To determine if this is a general trend across CTD2, GDSC, and PRISM drug datasets, we separated drug targets that are CRISPR pan-dependencies and found that the target dependency profiles were more often top drug correlates using RNAi (Fig. 3f). With the exception of CRISPR pan-dependencies, all other targets were more often top drug correlates using CRISPR (Fig. 3f). This suggests that RNAi and CRISPR datasets can be used to identify complementary sets of drug targets. It also indicates that the selectivity observed when using RNAi to knockdown CRISPR pan-dependencies can be recapitulated by small-molecule inhibitors and could potentially be used to identify therapeutically relevant targets.

### Functional relationships identified by co-dependency network

Large-scale genetic perturbation screens have been used to construct functional similarity networks and reveal novel gene-gene relationships [41–43]. A common method of constructing similarity networks from CRISPR or RNAi screens is to draw edges between pairs of genes that share a pattern of dependency across cell lines (co-dependencies). Using this approach, co-dependency results could differ between networks for a gene if the selectivity of its dependency profile differs greatly between datasets.

Based on our observations that CRISPR pan-dependencies often have stronger associations with omics markers and drug sensitivity using RNAi, we hypothesized that functionally related co-dependencies of CRISPR pan-dependencies would also be better resolved using RNAi. To test this hypothesis, we created a functional similarity network for each CRISPR and RNAi dataset and measured enrichment of previously established gene-gene relationships (CORUM, InWeb PPI, KEGG, Ensembl Paralog) among the top co-dependencies of each gene (Additional file 1: Fig. S17a). We found that for CRISPR pan-dependencies, the co-dependencies were more enriched for established related genes using the RNAi network, but all other co-dependencies were more enriched for related genes using the CRISPR network (Fig. 4a, b). This is consistent with previous observations that RNAi screens better identify functional relationships within essential protein complexes [42]. We also observed variation in the performance of CRISPR pan-dependencies between networks created from datasets of the same perturbation type, but to a lesser extent than the variation observed between RNAi and CRISPR networks (Additional file 1: Supplemental Notes, Additional file 1: Fig. S18). This suggests it could be beneficial to use both CRISPR and RNAi networks to identify functionally related genes.

Ideally, we would like to integrate the gene–gene networks from different perturbation types into a single network that could be used to study co-dependency relationships between all genes. As a preliminary estimate of whether CRISPR and RNAi networks can be integrated in a way that captures both shared and complementary information, we used a nonlinear integration method, Similarity Network Fusion [44], to create a CRISPR-RNAi fusion network (Additional file 1: Fig. S17b). Evaluating the co-dependencies of the fusion network, we found that it improved the enrichment of related genes for CRISPR pan-dependencies as compared to the CRISPR network alone (Fig. 4a). For example, ribosomal protein S21 (RPS21) is a CRISPR pan-dependency that has several other related ribosomal proteins as co-dependencies using RNAi (RPS27A, RPS16,

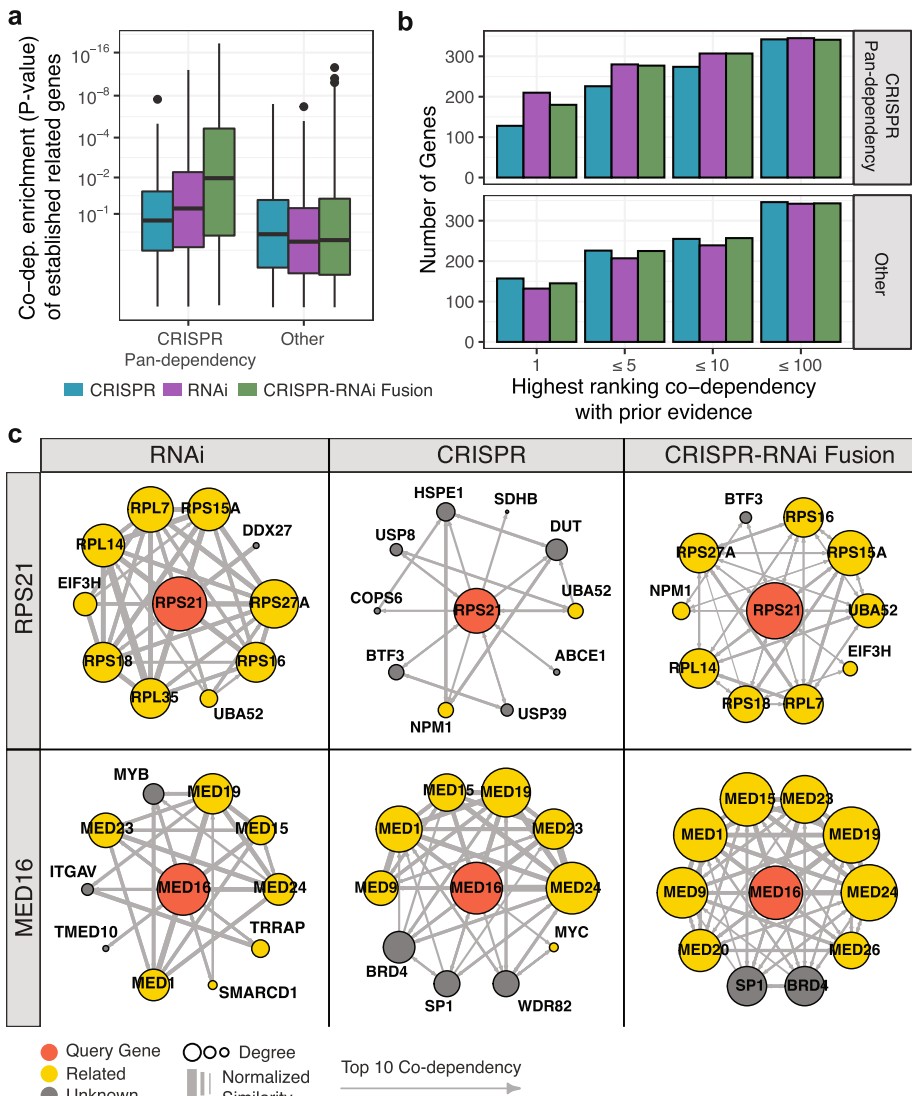

**Fig. 4** Co-dependencies in RNAi, CRISPR, and integrated datasets. **a** Co-dependencies of each query gene (high-confidence dependencies with at least 3 dependent cell lines in either CRISPR or RNAi dataset) were tested for enrichment of previously established related genes (CORUM, PPI, KEGG, paralogs). Query genes that were classified as CRISPR pan-dependencies are separated from the others. **b** Number of query genes for which one of the gene's related priors is among the query gene's top 1, 5, 10, or 100 ranked co-dependencies. Query genes are separated by CRISPR pan-dependency status as in part **b**. **c** Examples of local co-dependency networks for query genes, RPS21 (pan-dependency) and MED16 (selective dependency), constructed from CRISPR, RNAi, and integrated CRISPR-RNAi gene similarity networks. Only the top 10 co-dependencies of the query gene are included as vertices, but a directional link is shown between any two vertices if the target is a top 10 co-dependency of the source. Size of a vertex corresponds to the number of connections between the vertex and other gene vertices in the network. Edge weights are scaled respective to the full similarity matrix for each dataset

RPS15A, RPL7, RPS18, RPL14) that are retained in the fusion network, but not observed using the CRISPR network (Fig. 4c). Among the other genes, where the CRISPR network tends to have better performance (Fig. 4a), we also observed cases where the fused network was an improvement to RNAi alone (Fig. 4a). For example, mediator complex subunit 16 (MED16) is more tightly connected to other mediator complex subunits

using the fusion network compared to RNAi (Fig. 4c). Overall, the fusion network was less likely to have a related gene as the top ranked co-dependency when comparing to RNAi or CRISPR networks in their dominant gene classes (pan-dependency and other, respectively), but the fusion network was as likely as any other network to have a related gene in the top 10 co-dependencies across all dependency classes (Fig. 4b). This suggests that integrating networks derived from multiple experiments or perturbation types could help improve the consistency with which functional relationships are identified by co-dependencies.

## Discussion

Although mechanisms for cellular compensation to CRISPR gene knockout have been reported [45], we did not observe any widespread effects on cell viability. We found that dependencies detected with RNAi were typically a subset of CRISPR dependencies. In cases where genes exhibited stronger mean effects with RNAi, we were unable to rule out the possibility of technical artifacts (Additional file 1: Fig. S19). Most outliers with stronger RNAi effects, such as RBX1, were classified as pan-dependencies across all datasets, but demonstrated poor agreement on the mean gene effect between reagent libraries. Therefore, we determined that the CRISPR knockout screens were sufficient to produce comprehensive lists of dependencies for each cancer cell line.

The risk in exclusively using CRISPR knockout to screen cancer models for potential therapeutic targets is that many genes with pan-dependent profiles using CRISPR might not be considered viable targets if RNAi data was not available to reveal a potential therapeutic window. Based on clinical trials and FDA approvals, there is precedent to suggest these genes could indeed be used as targets for precision cancer therapies. For example, BRD4 (targeted by bromodomain inhibitors), CDC7 (TAK-931), HDAC3 (vorinostat), PRMT5 (GSK-3326595), NEDD8 (MLN4924), and ATR (VX-970), as well as XPO1 (selinexor), which is approved for advanced diffuse large B cell lymphoma and multiple myeloma [46], are all pan-dependencies using CRISPR knockout and selective dependencies using RNAi suppression. Given the downstream complexity of developing cancer therapeutics, it could be beneficial to cast a wider net at the target identification stage and include pan-dependencies that have mechanistic biomarkers for sensitivity to RNAi suppression, with an understanding that these targets might require chemotherapy-like treatment paradigms for successful clinical trial progression [46].

In the future, employing genetic screening with various gene inhibition approaches may offer significant benefits. Here we have shown that selective dependency profiles are the most useful for associating dependencies with other data types. Furthermore, the degree of inhibition required to elicit selective responses can vary for each gene. Therefore, using different levels of gene inhibition could enable the discovery of selective responses that might be missed when using RNAi or CRISPR knockout alone.

CRISPRi is a likely candidate because it could provide more control over the level of partial gene suppression while also maintaining the improved specificity of CRISPR reagents. However, CRISPRi reagents that are perfectly matched to the target sequence have very similar effects on cell viability as CRISPR-cas9 knockout [47] so partial suppression would likely require an attenuation of the reagents, such as the single base mismatch guides that have been shown to produce a range of activity from 0 to 100% of the

perfect match phenotype [48]. This strategy could be used to explore additional selectivity among the genes with pan-lethal response to CRISPR knockout.

While others have created an integrated RNAi and CRISPR dataset [49], we have only integrated datasets of the same perturbation type [16, 50]. This is because we expect screens of the same perturbation type, say CRISPR knockout, to produce concordant phenotypes for a cell line included in multiple experiments. In this case, data integration can help to reduce experimental noise and increase the sample size. On the other hand, different perturbation types (CRISPR, RNAi) provide complementary information that is more challenging to retain in an integrated format. Therefore, we chose to combine the results of analyses for biomarker identification and drug mechanism of action, instead of integrating the underlying CRISPR and RNAi datasets. In the case of co-dependency networks, it could be beneficial to have a single network that captures the functional relationships between as many genes as possible so we integrate the CRISPR and RNAi gene-gene networks.

## Conclusions

The transition from RNAi to CRISPR-Cas9 screening has improved reagent performance and enabled the discovery of many new cancer dependencies. RNAi datasets required more reagents per gene and substantial processing, including correction for off-target effects using DEMETER2, to approach the specificity of unprocessed CRISPR screens. Additionally, the CRISPR efficacy was substantially higher, which is likely a result of a stronger biological response to complete knockout. One drawback of generating completely null phenotypes using CRISPR was that it reduced the dynamic range of pan-dependencies. CRISPR knockout of any of these ~2000 genes consistently reduced cell viability across the entire pan-cancer collection of cell lines, whereas RNAi knockdown showed variation in dependency that could be predicted by omics features and better correlated to drug sensitivities or other functionally related co-dependencies. By including the appropriate combination of perturbation types in each analysis, it could be possible to take advantage of the improved precision of CRISPR-Cas gene editing, while also using partial gene suppression to capture selective patterns of dependency across a broader range of genes.

## Methods

### Separation of positive/negative controls using CRISPR and RNAi reagent datasets

Unscaled reagent $\log_2$ fold-change (LFC) for RNAi (Achilles [51], DRIVE [51]) and CRISPR (Avana [22], KY [52]) datasets were filtered for shared cell lines ($N = 62$) and gene targets ($N = 7595$). Achilles RNAi data includes 55 k and 98 k libraries so the genes in the smaller 55 k set were used in the overlap to ensure each cell line has measurements for all reported genes. The filtered reagent-level LFC datasets, remapped to use consistent DepMap cell line identifiers and Entrez gene identifiers, are included in the extended data. We calculated the separation between core essential genes [24] (positive control) and nonessential genes [24] (negative control) by strictly standardized mean difference (SSMD) for each cell line per reagent dataset. Each dataset has a differing number of reagents per gene so for each reagent dataset, all reagents targeting any of the genes in the positive or negative control set were used to compute SSMD.

### Correlation of reagents targeting the same gene

Reagent LFC datasets filtered for shared cell lines and genes were centered and scaled per cell line to reduce batch effects. For each gene, the Pearson correlation between all pairs of reagents targeting the gene were computed and collapsed to the median correlation coefficient per gene. Correlation is not expected between reagents targeting nonessential genes due to low variance. Therefore, we calculated the variance across cell lines for each gene by taking the max variance of reagents targeting each gene and converting it to a percentile per dataset. This allowed us to evaluate correlation between reagents per gene with respect to the variance in the reagents.

### Constructing an unbiased essential gene set for a positive control

A measure of viability screening performance is the separation between positive and negative control genes. However, the essential genes commonly used as positive controls are often derived from either RNAi or CRISPR screens, making them potentially biased towards gene dependencies that are best detected with one of those methods. For the purpose of comparing the performance of RNAi to CRISPR, we require a set of positive controls derived from alternative methods ("unbiased essential genes").

The alternative methods contributing to the unbiased essential genes include gene trap experiments using two human cell lines, KBM7 and HAP1, from Blomen et al. [25], embryonic lethal mouse genes identified by Dickinson et al. [26], and studies of human population-level exome and genome sequencing that suggest scores to quantify gene loss-of-function tolerance [27–32]. Since most of these studies provide a continuous metric for essentiality, creating a discrete gene set required cut-offs. For the human cell line data, we used genes that are overlapping hits in both cell lines (selected = "YES" in Supplemental Tables 1 and 2) [25]. For the human population studies, we convert the RVIS, pLI, Phi, missense *Z*-score, LoFtool, and $s_{het}$ metrics to percentiles and take genes with median percentile greater than 85, which Bartha et al. [53] suggest is similar to the commonly used cut-off of pLI > 0.9. For the mouse embryonic lethals, we use the genes that are indicated as lethal by MGI or IMPC (MGI_lethal = "Y", IMPC = "Lethal", or IMPC = "Subviable") in their Supplementary table 8 [26]. Since the CRISPR and RNAi experiments are conducted in vitro using cancer cell lines, we put more weight on the gene trap experiments since they also use human cell lines. For a gene to be included in the unbiased essential gene set, the gene must be included in the gene trap essential set, and be supported by one other source, either the mouse lethal or the human population lethal sets. This results in 801 genes. A summary of all metrics used in the classification scheme is included in the extended data.

### Classification accuracy (ROC AUC) of positive/negative controls using processed data

The unscaled reagent LFC datasets were collapsed to gene-level LFC values per cell line by simple average of all reagents targeting each gene. For comparison, the datasets processed using DEMETER2 (Achilles [51], DRIVE [51]) or CERES (Avana [54], KY [52]) are also unscaled. Processed and unprocessed mean LFC datasets were filtered to the 5226 genes and 62 cell lines that are shared between all datasets after mapping to DepMap cell line identifiers and Entrez gene IDs (provided in the extended data). To calculate the ROC AUC for each cell line, the unbiased essential gene set was used as the

positive class and non-expressed genes $(\log_2(\text{TPM}+1)<0.2$ using the DepMap 19Q1 RNAseq expression for protein coding genes) calculated per cell line were used as the negative class. The R package pROC (version 1.13.0) is used to build the ROC curves and compute the confidence interval of the curve.

### Correlation of matching genes across datasets of different perturbation types

Unprocessed (mean LFC) and processed (CERES and DEMETER2 gene effects) datasets are centered by the median of the nonessential genes and scaled by core essential genes [24]. For each pair of datasets where one is CRISPR and the other is RNAi, we calculated the Pearson correlation between matching genes ($N=5226$). This was done for the 4 cross-perturbation pairs (Achilles RNAi-KY CRISPR, Achilles RNAi-Avana CRISPR, DRIVE RNAi-Avana CRISPR, DRIVE RNAi-KY CRISPR) of unprocessed datasets as well as processed datasets. Since genes with very little variance in dependency are not expected to correlate between datasets, we binned the genes by the mean variance in the pair of datasets being evaluated.

Additionally, to determine the strength of each correlation relative to other correlations between the same pair of datasets, we compute the correlation between a gene and all other genes (5226) and then determine the rank of the correlation coefficient for the matching gene. Therefore, if a gene is given a rank of 1, it means that the most positively correlated gene in the opposite dataset is the matching gene.

### Pan-dependency analysis

Genes are ranked by gene effect within each cell line and divided by the total number of genes screened in each sample (varies due to missing values in some RNAi datasets). This produces a normalized rank between 0 and 1 where 0 is the most essential gene and 1 is the least essential (usually enriched). Then for each gene, the 90th percentile of the normalized ranks is calculated across cell lines, which represents the rank of the most dependent cell line among the 10% of least dependent cell lines. The distribution of 90th percentile ranks is bimodal, indicating there is a subset of genes that are a dependency in 90% of the cell lines, while the majority of genes are not. The minimum of the 90th percentile ranks distribution between the two peaks is used as the cut-off to separate the more essential mode, which defines the pan-dependency gene set. Varying the choice of percentile between the 80th, 90th, and 95th produces relatively stable pan-dependency lists [55].

### Dependency probabilities

To estimate the likelihood that a CERES or DEMETER2 gene effect value represents a true decrease in cell viability, we use a Bayesian inference method designed for Project Achilles and described as part of the Project Achilles data processing pipeline [55]. While others have developed similar methods [56] for identifying dependencies using log fold-change values, we wanted to calculate the probability of dependency for each gene effect value from CERES and DEMETER2, which is the primary use case for the Project Achilles inference method. For each cell line, the distribution of gene effects is decomposed into a null (non-expressed genes per cell line) and a positive control distribution (pan-dependent genes per dataset) and we calculate the probability that each

gene effect comes from the positive control distribution. All other parameters are Project Achilles defaults. In places where a discrete number of dependencies is provided per cell line, it is calculated by the number of genes with probability of dependency greater than 0.5. A non-dependency is a gene for which all cell lines have a probability of dependency less than 0.5.

**Non-dependency analysis**

Genes are labeled nonessential if the probability of dependency is less than 0.5 for all cell lines in the dataset.

**Defining a high-confidence dependency gene set across perturbation types**

To reduce outliers in dataset comparisons that are caused by experiment-specific details, such as reagent library design, timepoints, or cell line media, we defined a gene set for which datasets of the same perturbation type are in agreement, taking into consideration that the definition of agreement might be different for pan-dependencies, selective dependencies, or non-dependencies.

We used correlation as one measure of agreement between datasets. More specifically, we computed the Pearson correlation between all pairwise combinations of gene effect profiles between datasets, and then ranked the correlation between matching pairs of genes relative to all other non-matching pairwise combinations. However, correlation does not work well for pan-dependencies or non-dependencies that have little or no variance in CERES or DEMETER2 gene effects. Therefore, we also defined agreement as shared pan-dependencies or non-dependencies between datasets.

For the CRISPR datasets, 1339 genes were shared pan-dependencies, 6303 genes are not dependencies for any cell lines using either library, and 2324 genes are top correlates between libraries, resulting in a union of 9303 multi-library agreement genes out of the 17,001 genes that are shared targets in both Avana and KY libraries (54.5%). The RNAi libraries (DRIVE, Achilles) have 246 shared pan-dependencies, 725 top correlates, and 2327 shared non-dependencies, for which the union ($N=3067$) represents 44.5% of the 6894 shared targets between Achilles and DRIVE datasets since the DRIVE library is not genome-wide. We refer to the intersection of CRISPR dataset agreement and RNAi dataset agreement as the high-confidence dependency set ($N=1718$) and restrict comparisons to these genes when looking to highlight consistent differences between perturbation types. Genome-wide agreement metrics as well as the annotation for inclusion in the set are provided in Supplementary Table 1.

**Identifying strongly selective dependencies by likelihood ratio test**

Strongly selective dependencies are defined by a "likelihood ratio test", as described in McDonald et al. [4]. For each gene, the log-likelihood of the fit of CERES or DEMETER2 gene effect to a normal distribution and a skew-t distribution is computed using the R packages MASS and sn, respectively. In the event that the default fit to the skew-t distribution fails, a two-step fitting process is invoked. This involves keeping the degrees of freedom parameter (nu) fixed during an initial fit and then using the parameter estimates as starting values for a second fit without any fixed values. This process repeats up to 9 times using nu values in the list (2, 5, 10, 25, 50, 100, 250, 500, 1000) sequentially

until a solution is reached. The numerical optimization methods used for the estimates do not guarantee the maximum of the objective function is reached. The reported LRT score is calculated as follows:

$$LRT = 2^*[\ln(\text{likelihood for Skewed} - t) - \ln(\text{likelihood for Gaussian})]$$

A gene labeled as a strongly selective dependency (SSD) has an LRT greater than or equal to 100.

### High-variance dependencies

Variance in the probability of dependency across cell lines is calculated for each gene using RNAi and CRISPR data with matched cell lines ($N=403$) and gene targets ($N=15,221$). The use of dependency probabilities, as opposed to gene effects, ensures that genes with higher variance have cell counts that are more depleted, indicating a loss of cell fitness. High gene effect variance could indicate a cell proliferation phenotype.

A negative control set of non-expressed genes is constructed by calculating the density of the total distribution of RNAseq TPM (log2) for the 403 cell lines and 14,517 genes that overlap the dependency datasets. The density is bimodal with a minimum value between non-expressed and expressed modes that corresponds to a TPM (log2) value of 1.385. This value is used as a threshold for expression and genes that are below the threshold for all 403 cell lines are considered non-expressed ($N=643$). We then use the 99th percentile of the variance distribution for non-expressed genes as the cut-off for high-variance (0.038 for CRISPR, 0.0442 for RNAi).

### ISG signature

RNAseq TPM (log2) expression data is transformed into an ISG score per cell line using ssGSEA method implemented in the R package "gsva" version 1.30.0. Cell lines with hematopoietic and lymphoid tissue primary site are excluded. The gene set consists of the 38 genes provided by Liu et al. [57].

### Mapping dependency classes between RNAi and CRISPR

Using the shared cell lines and genes between the RNAi (D2-Combined) and CRISPR (Avana), each high-confidence dependency gene is mapped between the dependency classification using RNAi and classification using CRISPR dataset. The dependency class definitions are not mutually exclusive, e.g., high-variance dependencies can also be strongly selective, so to visualizing the mapping where each gene can only have one class we do the classification in the following order:

(1) Non-dependency: probability of dependency less than 0.5 for all cell lines
(2) Weakly selective: at least one dependent cell line
(3) Strongly selective dependency (SSD): LRT score greater than 100
(4) Pan-dependency: rank in 90th percentile least dependent cell line below threshold (defined per dataset)
(5) High-variance: variance in probability of dependency greater than threshold (defined per dataset)

(6) Strongly selective dependency (SSD): LRT > 100 and a negative skew (outlier cell lines have more negative gene effect)

## CRISPR and RNAi pan-dependencies supported by alternative methodologies

The three alternative methods of identifying essential genes used for comparison to CRISPR and RNAi are constraint of loss-of-function mutations in ExAC human population sequencing data, gene trap experiments on human cell lines, and mouse knockout lethality studies. Essential genes from ExAC were determined by ranking the median of RVIS, mis_z, pLI, phi, LofTool, and s_het scores [27–32] after converting to percentiles and taking the top 15%. Similarly, we also took the top 15% of genes ranked by the mean of gene trap percentiles for KBM7 and HAP1 cell lines [25]. Mouse knockout lethality is defined by Supplementary table 8 from Dickinson et al. [26] where the "MGI_lethal" column has a value of "Y" or "IMPC" column has a value of "Lethal" or "Subviable". The two sets of CRISPR or RNAi pan-dependencies are the result of the pan-dependency analyses performed on the matched CRISPR or RNAi gene effect datasets and filtered for high-confidence dependencies. The Venn diagrams are area-proportional Euler diagrams fit with the R package "eulerr" version 6.1.1.

## Gene set enrichment of distinct CRISPR and shared CRISPR-RNAi pan-dependencies

We separately tested enrichment of the distinct CRISPR pan-dependencies ($N = 208$) or the shared CRISPR-RNAi pan-dependencies ($N = 234$) for over representation of gene sets from the KEGG ($N = 186$), Biocarta ($N = 289$), Reactome ($N = 1499$), and GO ($N = 9996$) collections in MsigDB version 7.0. *P*-values and odds ratios were assigned by one-sided Fisher's exact test in the R package "stats" version 3.5.1 where the in-group is either shared or distinct pan-dependencies and the out-group is all other high-confidence dependencies. The *P*-values were adjusted for multiple hypothesis testing across all collections of gene sets using the Benjamini and Hochberg method implemented by the "p.adjust" function from the same package. All genes in the analysis are restricted to the high-confidence dependency set ($N = 1703$).

## Predicting CRISPR pan-dependency status using results from gene trap, RNAi, and ExAC

The target vector we are predicting is genes classified as pan-dependencies in both CRISPR datasets (Avana, KY) independently (positive class, $N = 1247$) and genes not classified as pan-dependencies in either CRISPR dataset (negative class, $N = 12,845$). We evaluate the accuracy of classifying the target vector by ROC AUC using a single predictive feature for each of the three methodologies (RNAi, gene trap, and ExAC). The mean D2-Combined gene effect across cell lines is used as the predictive feature representing RNAi ($N = 14,092$). The predictive feature for ExAC represents the constraint against loss-of-function in human population sequencing and is calculated by converting the RVIS, mis_z, pLI, phi, LofTool, and s_het scores to percentiles and taking the median for genes with values in at least 3 of the ExAC metrics. The *p*-value of gene trap experiments for KBM7 and HAP1 are converted to percentiles and the average percentile of each gene is used as the predictive feature ($N = 13,141$). The specificities, sensitivities, and ROC AUC are computed using the R package pROC version 1.13.0.

**Modeling pan-dependency status using mRNA and protein expression levels**

A binary target vector is defined such that 194 pan-dependencies unique to CRISPR are the positive class and the 225 pan-dependencies that are shared using RNAi or CRISPR are the negative class. A random forest classification model (R package "randomForest" version 4.6–14 with default parameters, including 500 trees and minimum terminal node size of 1) is fit to the two-class target vector where the predictive features included for each sample (perturbed target gene) are RNAseq or proteomic characterizations [58] of the target gene itself. Specifically, for each gene, we summarize the RNAseq expression (log2 TPM) and normalized protein abundance [59] distributions across 221 shared cell lines using the following metrics: mean, variance, 0.1 quantile, 0.9 quantile, fraction of cell lines where the gene is not detected, and a binary indicator of whether there were multiple isoforms of a gene detected. For proteins with multiple isoforms per gene, we use the descriptive statistics from the isoform that is most commonly detected. We also include the Pearson correlation between each gene's RNAseq expression and normalized protein abundance profile as a predictive feature.

To compare the utility of mRNA expression and protein features for predicting pan-dependency status, we trained three different models using (1) only protein features, (2) only mRNA features, or (3) all features (protein features, mRNA features, and the correlation feature). Predicted class probabilities are calculated using 5-fold stratified cross-validation and the accuracy is determined by the area under the receiver operating characteristic curve (ROC AUC). For each predictive feature, the R package "randomForestExplainer" version 0.10.1 was used to calculate the minimal depth of the variable across trees. The top 5 predictive features, according to lowest mean minimal depth, were used to fit a simplified tree to all pan-dependency classifications with the R package "rpart" version 4.1–15 to visualize an example of the interaction between features.

**Predicting each gene effect profile using molecular and cellular features**

Molecular and cell line annotation features were assembled into a large matrix of all predictive features, including datasets released by Cancer Dependency Map (RNAseq, relative copy number, damaging mutation, missense mutation, hotspot mutation, fusion, cell line tissue/disease type) and published CCLE datasets [58, 60] (RPPA, total proteomics, metabolomics, RRBS). Continuous features (RNAseq, relative copy number, RPPA, total proteomics, metabolomics, RRBS) were individually $z$-scored per feature and joined with one-hot encodings of categorical features (damaging mutation, missense mutation, hotspot mutation, fusion, cell line tissue/disease type). Cell lines without RNAseq data were dropped and any remaining missing values were assigned a zero. A confounder variable of the CRISPR or RNAi screens (SSMD – represents the separation between positive and negative controls) is also included as a predictive feature to control for technical aspects of screen quality.

Each gene effect profile from the CRISPR or RNAi datasets is modeled using two sets of predictive features. The first is the Related model where features are only selected if there is a prior known relationship between the perturbation target and the measured molecular feature suggested by InWeb protein-protein interaction network, CORUM, or paralogs based on DNA sequence similarity from Ensembl Compara (exception of

confounders and tissue/disease annotations that are always included). The second model is the Unbiased model where the top 1000 features according to Pearson correlation between feature and perturbation target are included without use of any other prior information. Random forest regression models (100 trees, max-depth of 8, and a minimum of 5 cell lines per leaf) from the Python scikit-learn package were trained using stratified 5-fold cross-validation. Once predictions were made for each held-out set, the correlation between predicted and observed gene effects was used as the accuracy per model. When a single predictive accuracy is presented for a gene, it represents the maximum of the accuracies for the Related and Unbiased models for the gene. Predictive feature importances are calculated for each model and are based on mean decrease in impurity (Gini Importance).

### Predictive marker classes

The top predictive features used by each accurate ($r > 0.5$) random forest model were annotated based on the relationship between the predictive features and the target gene. Dependencies are classified as genetic driver, expression addiction, synthetic lethal (paralog), CYCLOPS, oncogene addiction, or synthetic lethal (oncogene/tumor suppressor) if they meet any of the following criteria:

| Predictive marker class | Predictive accuracy > 0.5 | Feature relation to target | Feature importance | Feature type | Dep.-Feature Correlation direction | Dep. Skewness |
|---|---|---|---|---|---|---|
| Genetic driver | Any model | Self | Normalized feature importance > 0.05 | Hotspot or missense mutation, fusion, copy number | Negative (CN), Positive (Mut.) | Negative |
| Expression addiction | Model using self exp. feat | Self | Top feature of model | RNAseq | Negative | |
| Synthetic lethal (paralog) | Model using paralog as top feature | Other (paralog) | Top feature | Any | Any | |
| CYCLOPS | Model using self exp. feat | Self | Top feature on same chr. arm and self-feature has normalized feature importance > 0.05 | Copy number OR RNAseq | Positive cor. to top feature AND positive cor. to self CN | Negative |
| Oncogene addiction | Any model | Self (oncogene from oncoKB.org) | Top 10 of unbiased or related model | Any | Any | Negative |
| Synthetic lethal (oncogene/TSG) | Model using oncogene or TSG feature | Other (oncogene or TSG from oncoKB.org) | Normalized feature importance > 0.05 | Any | Any | Negative |

### Chemical and genetic screen correlation

Several large chemical screening datasets (GDSC [37, 38], CTD2 [6, 39], and PRISM [5, 40] secondary screen) were chosen based on cell line overlap with the genetic perturbation screens and availability of dose-level viability measurements. PRISM values represent log fold-change relative to DMSO, measured over an 8-point concentration range

after 5 days, corrected for experimental confounders using ComBat. While PRISM uses consistent media across cell lines and consistent concentrations across compounds, the CTD2 dataset (16-point concentration range with viability measured by CellTiter-Glo at 3-day endpoint) and GDSC dataset (9, 7, or 5 point concentration range with viability measured by Syto60, Resazurin, or CellTiter-Glo at 3-day endpoint) use provider-recommended media for each cell line and a range of concentrations selected specifically for each compound. GDSC2 data was selected over GDSC1 data when available.

All drug datasets were processed using DepMap identifiers for cell lines and Broad Institute IDs for compounds according to the mapping from The Drug Repurposing Hub (RepHub) [40]. Any remaining replicates were averaged and datasets were filtered to contain only compounds that have RepHub annotated gene targets included in the genetic perturbation datasets (including the Project DRIVE sub-genome library) and cell lines included in DepMap. Since the first principal component (PC1) has been previously reported [61] to be a correlate of growth rate using the GDSC datasets, we removed PC1 from each of the dose-level datasets using the removePrincipalComponents function from the R package "WGCNA" which returns the residuals after linear regression on PC1. Missing values were replaced with the median per drug dose prior to removing PC1 from the data, but these imputed values were removed for downstream analysis.

To compare drug-gene target correlation between CRISPR and RNAi datasets, we computed all pairwise Pearson correlations between dose-level drug perturbations (CTD2: [344 cell lines, 182 drugs, 16 doses], GDSC: [261 cell lines, 238 drugs, ~8 doses], PRISM: [251 cell lines, 978 drugs, 8 doses]) and the CRISPR or RNAi gene effect profiles for 3197 genes that were a dependency (probability of dependency > 0.5) for at least 3 cell lines using either CRISPR or RNAi. For each drug, we labeled its best correlated annotated target gene. For each annotated drug-gene target pair, we labeled the best correlated dose. For each comparison between CRISPR, RNAi, and an individual drug dataset, the cell lines, genes, and compounds are all matched. However, each drug dataset contains a different selection of compounds, cell lines, and doses, so these comparisons are not intended to identify meaningful differences between drug datasets.

### Co-dependency networks

RNAi and CRISPR datasets (matched for cell lines, genes, and missing values) were filtered for genes in the high-confidence dependency set which also have at least 3 dependent cell lines using CRISPR or RNAi ($N = 758$). For each genetic dependency dataset (CRISPR, RNAi), all pairwise gene-gene Pearson correlations were calculated to create a symmetric 758 gene by 758 gene matrix where higher values represent greater co-dependency. This simple method of generating a weighted adjacency matrix ("similarity network") is used as a baseline for individual CRISPR and RNAi co-dependency networks. To create an integrated CRISPR and RNAi similarity network, we used the algorithm (Additional file 1: Fig. S17b) for Similarity Network Fusion [44] (SNF) with and a few adjustments, most notably using Pearson correlation as the similarity metric instead of converting Euclidean distance to similarity (default). Sparse matrices were created using $k$ nearest neighbors ($k = 7$) with the value of $k$ determined by the log(number of columns) as suggested by the SNF authors. After 20 iterations, the nearly converged

CRISPR and RNAi weight matrices were averaged and the resulting CRISPR-RNAi fusion similarity matrix was used for comparisons to the baseline CRISPR and RNAi similarity matrices.

To compare the CRISPR-RNAi fusion network to the individual CRISPR and RNAi co-dependency networks, we evaluated each network based on the recovery of gene-gene relationships that are supported by prior evidence (PPI, CORUM, paralog, KEGG) that we refer to as "known related" genes. For each of the 758 genes included in the co-dependency networks, we performed a one-sided Kolmogorov-Smirnov test on the absolute values of the gene's similarity scores to all other genes in order to determine if its known related genes had higher scores than other genes. For each gene, we also determined its best ranking known related gene when sorted by absolute value of similarity score. When visualizing the local graph structure of an example query gene within the larger co-dependency network, we included edges between the query gene and its top 10 co-dependencies based on the absolute value of the similarity score. The weight of the edges corresponds to the *z*-score of the gene-gene similarity score where the mean and standard deviation of the entire similarity matrix is used to calculate the *z*-score.

## Supplementary Information

---

**Additional file 1.** Supplemental notes and figures.

**Additional file 2: Table S1.** High-confidence dependencies.

**Additional file 3: Table S2.** Gene dependency classes (pan-dependency, strongly selective dependency, etc.) and the related metrics.

**Additional file 4: Table S3.** Predictive models of dependency -- Accuracy and top features used to predict CRISPR and RNAi gene effects.

**Additional file 5.** Review history.

---

**Peer review information**

**Review history**
The review history is available as Additional file 5.

**Authors' contributions**
J.M.K., A.T., and T.R.G. conceived the work. J.M.K. performed the computational analyses, generated the figures, and wrote the manuscript. A.A.B. and B.R.P. contributed to analysis and/or interpretation of results. D.E.R. contributed to the acquisition of genome-wide CRISPR-Cas9 and RNAi screening data. J.M.D., J.S.B., W.C.H., J.M.M., F.V., and A.T. revised the manuscript. All authors approved the final manuscript.

**Funding**
This work was funded by The Robertson Foundation, and the Cancer Dependency Map Consortium. The funders did not influence the conception, design, or analysis of the study or the writing of this manuscript.

**Availability of data and materials**
The datasets generated and/or analyzed during the current study are available as a figshare collection (https://doi.org/10.6084/m9.figshare.c.5775152.v1). The "Raw data" archive (https://doi.org/10.6084/m9.figshare.16735132.v1) includes multi-omics data, drug and genetic perturbation data, and gene sets (essentiality controls [24, 53, 56], oncogenes/tumor suppressors from oncoKB.org [62], MSigDB version 7.0, InWeb PPI network, CORUM 3.0 human core complexes, DNA sequence paralogs from Ensembl Compara). These datasets have all been pre-processed to use consistent cell line, gene, and compound identifiers, for which the mappings and aliases have also been included. The provided multi-omics datasets are derived from published sources (metabolomics [60], RPPA [60], RRBS [60], proteomics [58]), produced for the Cancer Dependency Map 18Q4-20Q1 releases (RNAseq expression, relative copy number, and mutation), or computed directly (ssGSEA, EMT state, MSI) from one of the previously described omics datasets. The provided drug screening datasets are from published sources (PRISM Repurposing secondary screen [5], GDSC [37, 38], CTD2 [6, 39]) and include compound metadata from the Drug Repurposing Hub (https://clue.io/repurposing). The provided genetic perturbation

data is available as reagent log fold-change as well as unscaled or scaled DEMETER2 (RNAi) or CERES (CRISPR) gene effects. RNAi data originates from the figshare (https://doi.org/10.6084/m9.figshare.6025238.v4) for DEMETER2, which includes Project Achilles and Project DRIVE datasets. CRISPR data originates from the Cancer Dependency Map (Avana) 19Q1-20Q2 releases and the CERES processing of Project SCORE [3] (KY) data (https://doi.org/10.6084/m9.figshare.9116732.v2).

The results of each analysis are available from the "Processed data" archive (https://doi.org/10.6084/m9.figshare.17948639.v1). For genetic dependencies, this includes pan-dependency and selectivity (LRT) scores, probability of dependency, and detailed predictive modeling results (predicted gene effects and feature importance) for each dataset. For drug sensitivity, the viability datasets used to evaluate drug-gene target associations, which have PC1 removed, are included along with the correlation between all drugs and annotated targets. Co-dependency networks are provided in the form of weighted adjacency matrices. Other processed data files are described in more detail via Figshare.

Data analysis was performed using R and Python scripts that are available under the BSD-3-Clause license from GitHub (https://github.com/broadinstitute/depmap-crispr-vs-rnai) [63]. The code release used to generate the presented figures and tables is deposited in a Zenodo repository (http://doi.org/10.1109/5.771073) [64]. Full analyses can be reproduced using Snakemake pipelines with raw data inputs from Figshare, or individual scripts can be used with intermediate data products that are also available via Figshare. The code base for this project includes several functions, such as the skewed-t likelihood ratio test (LRT), that could be reused in the analyses of other large datasets.

## Declarations

### Ethics approval and consent to participate
Not applicable.

### Consent for publication
Not applicable.

### Competing interests
B.R.P. and F.V. receive research support from Novo Ventures. D.E.R. receives research funding from members of the Functional Genomics Consortium (Abbvie, BMS, Jannsen, Merck, Vir), and is a director of Addgene. T.R.G. receives research funding from Bayer HealthCare, Calico Life Sciences, and Novo Ventures, is a founder and equity holder of Sherlock Biosciences and FORMA Therapeutics, serves a consultant for GlaxoSmithKline, and was formerly a consultant and equity holder in Foundation Medicine acquired by Roche. W.C.H. is a consultant for Thermo Fisher, Solasta Ventures, MPM Capital, KSQ Therapeutics, iTeos, Tyra Biosciences, Frontier Medicine, Jubilant Therapeutics, RAPTTA Therapeutics, Function Oncology, and Calyx. A.T. is a consultant for Cedilla Therapeutics, Foghorn Therapeutics, The Center for Protein Degradation (Deerfield), Tango Therapeutics, and is an SAB member and holds equity in Turbine Simulated Cell Technologies. This work was partially funded by the Cancer Dependency Map Consortium, but no consortium member was involved in or influenced this study.

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

## 
