## [**Additional file 5.** Review history. · Genome Biology]

Review History

First round of review

Reviewer 1

Are you able to assess all statistics in the manuscript, including the appropriateness of statistical tests used? Yes, and I have assessed the statistics in my report.

Comments to author:

CRISPR knockout is intended to produce a complete loss of function in a target gene, whereas drugs or other targeted agents typically induce only a partial loss of function. While CRISPRko produces a robust association between complete KO and background-specific vulnerability, it may not be the ideal model for drug sensitivity given these different modalities. Krill-Burger et al compare CRISPRko complete knockout phenotypes with RNAi induced knockdown phenotypes, which presumably result from partial loss of fitness, and compare their profiles using several approaches, including direct comparison of fitness profiles, correlation/covariation with drug sensitivity and genomic/biomarker presence, and the extraction of gene functional information from correlated fitness profiles.

The key takeaway from this rather comprehensive study is that genes classified as pan-dependencies (genes essential in 90% of cell lines) by CRISPR tend not to have much dynamic range in their knockout phenotype, which renders them uninformative in comparison with other data sets. However, the RNAi approach does find some variation in knockdown phenotype, with substantial confidence added by the high proportion of CYCLOPS genes recovered using this approach. This variation is then useful in comparisons using, e.g., coessentiality and drug response, and buttresses one of the few weak spots in the large-scale CRISPR knockout screening efforts in cancer cell lines.

Overall this paper is exceptionally well written, the conclusions are supported by the data, and no overly ambitious conclusions are presented. I have no suggestions for improvement.

Reviewer 2

Are you able to assess all statistics in the manuscript, including the appropriateness of statistical tests used? No, I do not feel adequately qualified to assess the statistics.

Comments to author:

As the accumulation of large-scale genetic perturbation data, deeper understanding of molecular mechanisms behind complexed bioprocesses and accelerated identification of therapeutic targets may be achieved with established tools such as DepMap. The manuscript entitled "Partial gene suppression improves identification of cancer vulnerabilities when CRISPR-Cas9 knockout is pan-lethal" utilized large amount of genetic perturbation data to compare the technical difference between CRISPR knockout and RNAi in deciphering cell dependency genes. By direct comparison of different categories of gene dependencies, biomarker analysis, association with

drug sensitivity, and constructing co-dependency network, the authors confirmed previous findings that CRISPR methods are more robust than RNAi and identified more confident gene dependencies. More importantly, they revealed that the genes identified by RNAi with variations while showed pan-essential by CRISPR may contain functional targets of therapeutic value. Such targets may be overlooked because of its pan-essentiality and lack of selectivity if only focusing on CRISPR data.

These findings are interesting as the action of the most targeted drugs tends to "partially" inhibit the function of its target, which may be better recapitulated by RNAi with partial suppression but by CRISPR with completely functional removal or deletion. By focusing on those RNAi-identified CRISPR pan-dependencies, it is possible to accelerate the development of promising targeted drugs for cancer therapy. The authors performed a series of analysis to corroborate this conclusion, however, there are still some concerns or suggestions for this manuscript.

1. To side-by-side compare the two technical routes, the authors put all the efforts on the negative selection side for the dependencies which are the central aim of such large-scale genetic perturbation project. The decreased cell viability phenotype can be affected by direct perturbation as well as many indirect effects unrelated to target gene inhibition. For a thorough comparison between RNAi and CRISPR, did the authors try to take a look at positive selection side?
2. The authors showed that a large amount of high variance and selective dependency by RNAi are CRISPR pan-dependent and this is possibly due to partial inhibition from RNAi. How to explain those high variance and selective dependency by CRISPR? To which extent that they are truly selective or varied due to technical flaws?
3. In addition to some of the CRISPR pan-dependency, those truly selective dependencies are important and may also serve as potential therapeutic targets. It will be great that the authors can provide a tentative list for such targets and also display top predictive biomarkers to inform their selective dependency if possible.
4. Please provide statistics for Fig. 2d and 2f.
5. For comparison of mean gene effect of CRISPR and RNAi (Supplementary Fig. 19), a few of RNAi stronger dependency were observed. Did those genes show absolutely non or weak negative selection by CRISPR? This may be due to inefficient sgRNA design for the specific CRISPR library, but not the technical flaws of CRISPR technology. This issue should be discussed.
6. The key thing for these discrepancies between RNAi screen and CRISPR screen seems to be "partial inhibition vs. complete inhibition". Is it possible that the authors take a brief analysis of limited CRISPRi screen data, which is also partial inhibition for perturbed genes, to see whether the main findings still hold? These efforts may guide future directions for dependency screens.

Reviewer comment #1: To side-by-side compare the two technical routes, the authors put all the efforts on the negative selection side for the dependencies which are the central aim of such large-scale genetic perturbation project. The decreased cell viability phenotype can be affected by direct perturbation as well as many indirect effects unrelated to target gene inhibition. For a thorough comparison between RNAi and CRISPR, did the authors try to take a look at positive selection side?

We appreciate the reviewer's suggestion to consider the positive selection side for a thorough comparison between RNAi and CRISPR. However, measuring positive selection in our functional genomic screens presents several challenges. Firstly, cell proliferation measurements tend to have more noise compared to cell dropout measurements. For example, many oncogenes have correlated CRISPR profiles between the Broad and Sanger datasets, but TP53 is the only highly correlated tumor suppressor (Positive Selection Fig. 1a, below). Secondly, there is a lack of suitable controls for normalizing across screens, as there are no genes known to consistently increase proliferation across all cell lines in our collection. This is likely because cancer cell lines have undergone a selection process to increase cell growth whereby tumor suppressor loss has already occurred in the models that would benefit from it.

Despite these challenges, we have analyzed the positive selection side to some extent. We previously observed that tumor suppressor genes (TSGs) are significantly enriched among the skewed CRISPR or RNAi gene effect distributions (Supplemental Fig. 6B). A key difference being that oncogenes and TSG tend to have left-skewed and right-skewed distributions, respectively (Positive Selection Fig. 1b). Furthermore, we compared gene effects for each TSG between cell lines that are wild-type and those with a prior loss (damaging mutation or copy number loss) of the corresponding TSG (Positive Selection Fig. 1c) to determine if TSG loss in the wild-type cell lines has a proliferative effect. This strategy allowed us to identify TP53 as a TSG using both CRISPR and RNAi data, while PTEN could only be identified as a TSG using CRISPR (Positive Selection Fig. 1c,d).

However, due to the limited number of detectable TSGs, it is difficult to draw any overall trends specific to CRISPR or RNAi in the context of positive selection. We acknowledge that a more comprehensive analysis of positive selection would be valuable, but the limitations and challenges mentioned above restrict our ability to make a thorough comparison between RNAi and CRISPR in this aspect.

Positive Selection Figure 1

a, Pearson correlation between the CRISPR gene effects from the Broad (Avena) and Sanger (KY) datasets. Genes are grouped by Oncology Knowledge Base (OncoKB) classification of oncogene or tumor suppressor gene (TSG). The genes shown (N=36) are those that have some selective signal (LRT > 100) for which correlation might be identified. **b**, Density of gene effects from the Broad CRISPR dataset for an oncogene (BRAF) and TSG (TP53). **c**, Two class comparison between cell lines that are wild-type for a TSG versus those that have complete loss (multiple damaging mutations, deep deletion, or copy number loss and a single damaging mutation) of the TSG. A one-sided t-test is used to determine if the TSG gene effects are more positive in the wild-type group. The 124 genes tested were listed as TSGs by OncoKB and have at least 3 cell lines that are wild-type and 3 cell lines that have complete loss of the TSG. The P-values are adjusted using Benjamini & Hochberg and the resulting FDR for CRISPR and RNAi comparisons are plotted against each other. **d**, Gene effects for PTEN and TP53 grouped by perturbation type and TSG status (complete loss, wild-type) that were tested for significance in part c.

Reviewer comment #2: The authors showed that a large amount of high variance and selective dependency by RNAi are CRISPR pan-dependent and this is possibly due to partial inhibition from RNAi. How to explain those high variance and selective dependency by CRISPR? To which extent that they are truly selective or varied due to technical flaws?

As shown in Supplemental Figure 7, strongly selective CRISPR profiles can occur when a subset of cell lines have a mutation that makes them vulnerable to a gene knockout that other cells can tolerate, e.g. BRAF (Supplemental Fig. 7a). High variance CRISPR profiles can occur when cell states cycle or change and create heterogeneity within the cell population, resulting in a cell-specific response to knockout (Supplemental Fig. 7b). Alternatively, as described in the manuscript (lines 132-137), variance in measured cell counts could arise from perturbations that slow growth rate to varying degrees rather than cause cell death.

To distinguish between CRISPR selectivity that is driven by biological factors, such as those listed above, as opposed to technical flaws, we look for correlation between a gene's dependency profile in different CRISPR datasets (Supplemental Figure 3, Supplemental Table 1) or selective dependency profiles that can be predicted by cell line omics features (Supplemental Figure 14, Supplemental Table 3). Comparisons between CRISPR and RNAi were already filtered to only include genes that agree between two datasets of the same perturbation type (Achilles/DRIVE, Avena/KY) to limit library-specific reagent issues. Additional high confidence CRISPR profiles beyond the comparisons could be identified using the 'CRISPR predictive accuracy' column from Supplemental Table 3 and filtering out cases where the 'CRISPR top predictive feature' column includes the term "Confounders".

Reviewer comment #3: In addition to some of the CRISPR pan-dependency, those truly selective dependencies are important and may also serve as potential therapeutic targets. It will be great that the authors can provide a tentative list for such targets and also display top predictive biomarkers to inform their selective dependency if possible.

The strongly selective CRISPR dependencies are indicated in Supplemental Table 2 by the column 'CRISPR Strongly Selective Dependency'. The summarized CRISPR predictive marker information can be found in Supplemental Table 3 and the complete results are available on Figshare (<https://doi.org/10.6084/m9.figshare.17948639.v1>), which includes the main results from our genome-wide multivariate predictive analysis described above. The column 'CRISPR predictive accuracy' is the correlation between the observed (measured) and predicted gene effects. The top predictive feature of each model (according to the Gini importance – mean decrease in variance at each split using the feature) is also included in the column 'CRISPR top predictive feature' and its corresponding normalized Gini importance (importance of all features used by the model sum to 1) in the column 'CRISPR top feature importance'.

Since most target discovery analyses will likely be beyond the scope of the data included in this CRISPR/RNAi comparison, the DepMap portal (<https://depmap.org>) is a better resource for predictive modeling results because it is regularly updated (bi-annually) with additional CRISPR

screens and omics features. The predictive modeling results, using the same core methodology described here, can be downloaded from the Cancer Dependency Map portal (https://depmap.org/portal/gene/predictability_files).

Reviewer comment #4: Please provide statistics for Fig. 2d and 2f.

Thank you for pointing this out. Figures 2d,f have been updated with Wilcoxon rank sum test (also known as Mann-Whitney test).

Figure 2 | Predictive models for dependency

a, Multivariate regression of each CRISPR or RNAi gene effect profile (N=15,221) using predictive features derived from omics datasets and cell line annotations. The accuracy of each predictive model is the correlation coefficient of measured and predicted values across cell lines. The number of accurate predictive models ($r > 0.5$) is split according to whether the gene target is also a CRISPR pan-dependency. **b**, Mean predictive accuracy for high-confidence dependencies (N=1,703) as a function of the fraction of cell lines identified as dependencies using CRISPR (probability of dependency > 0.5). **c**, CYCLOPS genes have a positive correlation ($r > 0.5$) between RNAi gene effects and copy number and account for ~35% of the accurate RNAi models for CRISPR pan-dependencies (N=1,719) after removing 148 RNAi models driven by confounding factors. **d**, Genetic perturbation of PRMT5 using RNAi results in larger effect size between cell lines with MTAP copy-number loss and MTAP wild-type compared to CRISPR knockout (Wilcoxon p-value: CRISPR = 4.6×10^{-8} , RNAi = 2.5×10^{-34}). **e**, RNAi gene effect of RBBP4 is more correlated with expression of paralog gene RBBP7. Density (2D)

contours represent 402 cell lines for each genetic dependency dataset (RNAi, CRISPR). **f**, CRISPR knockout of RAB6A has a more negative viability effect on cells lacking expression of paralog gene RAB6B (Wilcoxon p-value: CRISPR = 6.2×10^{-24} , RNAi = 6.1×10^{-7}).

Reviewer comment #5: For comparison of mean gene effect of CRISPR and RNAi (Supplementary Fig. 19), a few of RNAi stronger dependency were observed. Did those genes show absolutely non or weak negative selection by CRISPR? This may be due to inefficient sgRNA design for the specific CRISPR library, but not the technical flaws of CRISPR technology. This issue should be discussed.

The major trend observed in this study is that CRISPR knockout has a stronger and more consistent effect on cell viability than RNAi knockdown (Fig. 1a). Given this observation, the handful of outlier genes observed to have more negative mean RNAi gene effects are most likely caused by technical characteristics of the CRISPR or RNAi reagents chosen to target these particular genes, but they need to be evaluated on a case by case basis. In general, we tried to limit the impact of library-specific design decisions by requiring correlation between datasets of the same perturbation type (CRISPR: Avana/KY, RNAi: Achilles/DRIVE), but we had to make an exception for pan-lethal genes because their signal was too saturated to correlate well between datasets (Supplemental Figure 3). Therefore, we considered genes to be in agreement between datasets if the gene was classified as a pan-dependency in both datasets, which was sufficient for our primary analyses aimed at identifying associations between dependencies and other datasets. However, comparing mean gene effect between datasets highlighted some edge cases in the agreement scheme, such as genes that are pan-dependencies in all datasets, but differ in the degree of negative mean gene effect.

Upon closer inspection of the outliers with significantly more negative RNAi gene effects from Supplemental Figure 19a, we could not find any evidence to suggest that stronger RNAi effects are caused by other factors beyond technical artifacts of the experiment or reagent design. The pan-dependencies with the greatest difference in mean between CRISPR and RNAi change depending on the CRISPR library used in the comparison (Supplemental Fig. 19 b,c). This variance in the mean of essential genes was previously described in the comparison of Broad/Sanger CRISPR datasets (Dempster et al., 2019, Nat. Comm. DOI). There were a few genes that had more negative RNAi means when comparing to either CRISPR dataset, but the RNAi effect is driven by a small subset of DRIVE reagents (Supplemental Fig. 19d). For the 3 outliers that were not pan-dependencies (ASH2L, RNF125, FUBP1), only RNF125 had better correlation between RNAi datasets (Supplemental Fig. 19e) and this might be trivial because the agreement was between the same 4 shRNAs that were used in both RNAi libraries while the shRNAs that are distinct to each library are highly variable (Supplemental Fig. 19g).

We have updated Supplemental Figure 19 and the corresponding discussion paragraph to provide more information about the type of dependency profile observed for the outliers and the potential for reagent-specific effects (shown below).

Discussion (line #447):

Although mechanisms for cellular compensation to CRISPR gene knockout have been reported⁴⁵, we did not observe any widespread effects on cell viability. We found that dependencies detected with RNAi were typically a subset of CRISPR dependencies. In cases where genes exhibited stronger mean effects with RNAi, we were unable to rule out the possibility of technical artifacts (Supplemental Fig. 19). Most outliers with stronger RNAi effects, such as RBX1, were classified as pan-dependencies across all datasets, but demonstrated poor agreement on the mean gene effect between reagent libraries. Therefore, we determined that the CRISPR knockout screens were sufficient to produce comprehensive lists of dependencies for each cancer cell line.

Supplemental Fig. 19 | Comparison of mean gene effect using CRISPR and RNAi

a, Wilcoxon test between RNAi and CRISPR dependency scores (probability of dependency) for each gene with a high confidence dependency class that is also a dependency in at least one cell line (N=800). **b**, Comparison of the mean gene effect across cell lines using the CRISPR (Avana library) and the RNAi (DEMETER2 combined Achilles & DRIVE). These are the same datasets used in part (a) and the labeled genes correspond to the genes found to have stronger RNAi dependency. **c**, Many of the pan-dependent outliers with more negative mean RNAi effect from part (b) are no longer outliers when the CRISPR dataset is switched to the Sanger Institute KY library, suggesting the mean gene effect is not stable enough to conclude there are differences in knockout versus knockdown. **d**, Example of 2 genes (RBX1, USP39) that are pan-dependencies in all 4 datasets, have stronger RNAi dependency as shown in part (a). Points represent all individual reagents used to target the genes. Reagents are summarized across cell lines by mean log fold-change after a per cell line normalization (z-score). There is a subset of DRIVE reagents that may contribute to the overall more negative RNAi gene effect estimate from DEMETER2, but the consensus of RNAi reagents is not more negative than the CRISPR datasets. **e**, Correlation between datasets of the same perturbation modality for the 3 genes (RNF125, FUBP1, ASH2L) that have more negative RNAi gene effects, but are not pan-dependencies. **f**, Accuracy of multivariate regression models of gene effects using omics profiles and cell line features as predictors. The top omics predictors (Gini importance) for FUBP1 and ASH2L dependency are copy number features, MLLT1 and ASH2L respectively. **g**, Example of a selective dependency (RNF125) from part (a) that is better correlated between RNAi datasets in part (e). Reagents are summarized as in part (d) except the groups are further separated into reagents that are shared or unique between libraries. Unique RNAi reagents for RNF125 are highly variable and CRISPR reagents do not indicate an effect on cell viability.

6. The key thing for these discrepancies between RNAi screen and CRISPR screen seems to be "partial inhibition vs. complete inhibition". Is it possible that the authors take a brief analysis of limited CRISPRi screen data, which is also partial inhibition for perturbed genes, to see whether the main findings still hold? These efforts may guide future directions for dependency screens.

We agree with the reviewer's comment that it would be beneficial to include CRISPRi in the comparison, but the relevant data are not available yet. Currently, the genome-wide CRISPRi experiment (Sanson et al. 2018 Nat. Comm. DOI) with the most overlap in cell lines with the DepMap datasets (A375, HT29) used sgRNAs that are a perfect match to the target sequence. This resulted in levels of gene inhibition that produced a very similar phenotype as CRISPR knockout (CRISPRi Fig. 1) as noted by the authors. The CRISPRi reagent design methodology that is more relevant to this study is the systematic attenuation of CRISPR guide RNAs from Jost et al. (Nat. Biotech., 2020, DOI) where a series of single mismatch guides were used for each gene target and their relative activity compared to a perfect match reagent. Unfortunately, Jost et al. used three cell lines (K562, HeLa, Jurkat) for which only K562 overlaps the DepMap datasets and for that screen there were only 2,406 genes targeted with CRISPRi of which only 1,336 genes overlap with the RNAi sub-genome library that was used for K562. It might have been possible to look for a contrast between perfect match and mismatch reagents in their correlation to CRISPR-ko or RNAi screens, but with such a small sample, even the perfect

match reagents that Jost et al. used as the baseline do not strongly correlate with DepMap datasets. Therefore, it's too difficult to draw any conclusions from the mismatch CRISPRi reagents when there are too few genes to establish a solid baseline.

We have updated the discussion with additional details about the type of CRISPRi data that would be potentially useful for further interrogating the pan-lethal knockouts beyond what we were able to do with RNAi.

Discussion (line #469):

In the future, employing genetic screening with various gene inhibition approaches may offer significant benefits. Here we have shown that selective dependency profiles are the most useful for associating dependencies with other data types. Furthermore, the degree of inhibition required to elicit selective responses can vary for each gene. Therefore, using different levels of gene inhibition could enable the discovery of selective responses that might be missed when using RNAi or CRISPR knockout alone.

CRISPRi is a likely candidate because it could provide more control over the level of partial gene suppression while also maintaining the improved specificity of CRISPR reagents. However, CRISPRi reagents that are perfectly matched to the target sequence have very similar effects on cell viability as CRISPR-cas9 knockout (Doench 2018) so partial suppression would likely require an attenuation of the reagents, such as the single base mismatch guides that have been shown to produce a range of activity from 0-100% of the perfect match phenotype (Weissman 2020). This strategy could be used to explore additional selectivity among the genes with pan-lethal response to CRISPR knockout.

a

CRISPRi Figure 1

a, Correlation across genes for each cell line between the different combinations of perturbation

modalities (CRISPR knockout: C_{KO} , CRISPRi: C_i , RNAi: R_i). The facets indicate which experiment the CRISPRi data is from, while the CRISPR knockout and RNAi screens are from DepMap. The experiment from Sanson et al. has 2 overlapping cell lines (A375, HT29) and 10,953 overlapping genes. The Jost et al. experiment has 1 overlapping cell line (K562) and 1,336 genes.

Second round of review

Reviewer 2

The authors have addressed all the concerns and the manuscript is ready for acceptance.